# The Neurochemistry of Autism

**DOI:** 10.3390/brainsci10030163

**Published:** 2020-03-13

**Authors:** Rosa Marotta, Maria C. Risoleo, Giovanni Messina, Lucia Parisi, Marco Carotenuto, Luigi Vetri, Michele Roccella

**Affiliations:** 1Department of Medical and Surgical Sciences, University "Magna Graecia", Catanzaro 88100, Italy; marotta@unicz.it (R.M.); mariacristinarisoleo@yahoo.it (M.C.R.); 2Clinic of Child and Adolescent Neuropsychiatry, Department of Mental Health, Physical and Preventive Medicine, University of Campania “Luigi Vanvitelli”, Napoli 80138, Italy; marco.carotenuto@unicampania.it; 3Department of Clinical and Experimental Medicine, University of Foggia, Foggia 71100, Italy; giovanni.messina@unifg.it; 4Department of Psychology, Educational and Science and Human Movement, University of Palermo, Palermo 90128, Italy; lucia.parisi@unipa.it (L.P.); michele.roccella@unipa.it (M.R.); 5Department of Sciences for Health Promotion and Mother and Child Care “G. D’Alessandro”, University of Palermo, Palermo 90127, Italy

**Keywords:** autism spectrum disorder, neurochemistry, GABA, glutamate, serotonin, dopamine, acetylcholine, N-acetyl aspartate, oxytocin, melatonin

## Abstract

Autism spectrum disorder (ASD) refers to complex neurobehavioral and neurodevelopmental conditions characterized by impaired social interaction and communication, restricted and repetitive patterns of behavior or interests, and altered sensory processing. Environmental, immunological, genetic, and epigenetic factors are implicated in the pathophysiology of autism and provoke the occurrence of neuroanatomical and neurochemical events relatively early in the development of the central nervous system. Many neurochemical pathways are involved in determining ASD; however, how these complex networks interact and cause the onset of the core symptoms of autism remains unclear. Further studies on neurochemical alterations in autism are necessary to clarify the early neurodevelopmental variations behind the enormous heterogeneity of autism spectrum disorder, and therefore lead to new approaches for the treatment and prevention of autism. In this review, we aim to delineate the state-of-the-art main research findings about the neurochemical alterations in autism etiology, and focuses on gamma aminobutyric acid (GABA) and glutamate, serotonin, dopamine, N-acetyl aspartate, oxytocin and arginine-vasopressin, melatonin, vitamin D, orexin, endogenous opioids, and acetylcholine. We also aim to suggest a possible related therapeutic approach that could improve the quality of ASD interventions. Over one hundred references were collected through electronic database searching in Medline and EMBASE (Ovid), Scopus (Elsevier), ERIC (Proquest), PubMed, and the Web of Science (ISI).

## 1. Introduction

Autism spectrum disorder (ASD) refers to complex neurobehavioral and neurodevelopmental conditions characterized by impaired social interaction and communication, restricted and repetitive patterns of behavior or interests, and altered sensory processing [1]. The prevalence of autism has significantly increased during the last two decades from two to five per 10,000 children to 1:59 children (one in 37 boys and one in 151 girls), and the prevalence in males is four times greater than females [2]. 

Increasing evidence underlines the biological basis of autism. In fact, onset symptoms are observed before three years of age and, in most cases, changes in social behavior or other slight autistic features are noticed in the first few months of life [3]. This suggests that behind the pathophysiology of autism there are neuroanatomical and neurochemical events occurring relatively early in the development of the central nervous system (CNS). Numerous studies have also shown that autism can often be comorbid with other neurological and psychiatric disorders, such as global development delay and cognitive deficits, epilepsy or electroencephalographic (EEG) anomalies, sleep disorders, developmental coordination disorder, neuropathies, Tourette syndrome, anxiety, oppositional defiant disorder, conduct disorder, attention deficit hyperactivity disorder (ADHD), mood disorders, psychosis, personality disorder, post-traumatic stress disorder, eating disorders, gender dysphoria, and substance abuse [4,5]. Moreover, there are several medical conditions comorbid to autism such as immunological disorders, gastrointestinal diseases, sleep-related breathing disorders, and there are several genetic syndromes commonly associated with autism (fragile X syndrome, Rett syndrome, Angelman syndrome, tuberous sclerosis complex, Phelan McDermid syndrome, Timothy syndrome, neurofibromatosis type 1, etc.) [6,7,8].

All these factors contribute to a phenotypic heterogeneity that necessarily reflects a complex multifactorial etiology of ASD. This has led most researchers to consider autism dimensionally rather than using a categorial approach.

To a large extent, the ASD etiopathogenesis is unknown. It is a multifactorial condition caused by both genetic and environmental factors. Moreover, it has become clear that autism has an important genetic component. Siblings of individuals with autism have a prevalence of 2.9% to 3.7%, which represents a nearly 100-fold increased risk as compared with the general population [9,10]. Twin studies have found concordance rates of 36% to 91% between monozygotic twins, and concordance rates of 1% between dizygotic twins [11].

The first data about the involvement of neurotransmission in autism were obtained several decades ago with studies on postmortem brain and measurements of bodily fluids, and, more recently, through molecular imaging and genetic evidence about neurotransmitters.

Neurotransmitters and neuropeptides play a fundamental role in normal brain development and contribute to memory, behavior, and motor activity regulation [12]. Indeed, they influence neuronal cell migration, differentiation, synaptogenesis, apoptosis, and synaptic pruning. Therefore, a neurotransmitter system dysfunction can lead to impairments in the processes of brain development, determining autism [13].

This review focuses on evidence that suggests a role for neurotransmission dysregulation in autism and how these alterations could be useful for pharmacologic intervention in autism or as precocious biomarkers.

## 2. Aims and Methods

All of the aforementioned reasons have led researchers to rethink their efforts to understand the neurochemical alterations underlying ASD. The aim of the current review was to collect an overview of original articles about the contribution of neurotransmitters and neuropeptides to the pathophysiology of autism with a focus on gamma aminobutyric acid (GABA) and glutamate, serotonin, dopamine, N-acetyl aspartate, oxytocin and arginine-vasopressin, melatonin, vitamin D, orexin, endogenous opioids, and acetylcholine.

This review helps to better delineate the state-of-the-art main research findings about the neurochemical alterations in autism etiology and suggests possible related therapeutic approaches that could improve the quality of ASD interventions.

To this end, over one hundred articles, published over the years, were reviewed by performing a search using the following syntax (autism or autism spectrum disorder or Asperger syndrome or pervasive developmental disorders (Title/Abstract)) and (GABA or glutamate or serotonin or dopamine or N-acetyl aspartate or oxytocin or arginine-vasopressin or melatonin or vitamin D or orexin or opioids or acetylcholine (Title/Abstract)). References were identified through electronic database searching in Medline (Ovid, 1946 to present) and EMBASE (Ovid), and they were adapted for Scopus (Elsevier), ERIC (Proquest), PubMed, and the Web of Science (ISI). The final database search was run on February 2020.

## 3. Gamma Aminobutyric Acid

Gamma aminobutyric acid (GABA) is derived from glutamate thanks to the action of glutamate decarboxylase and it has a complex and homeostatic relationship balancing neuronal excitability. In immature brains, GABA receptors are different from those of adult brains. GABA represents the main excitatory neurotransmitter during brain development, and it influences proliferation, migration, synapse maturation, differentiation, and cell death [14].

Alterations in gabaminergic and glutaminergic systems cause a disrupted excitatory/inhibitory balance and are also potential mechanisms for autistic behaviors and for various neurodevelopmental disorders.

The excitatory/inhibitory imbalance theory for social behavior impairments has been demonstrated through the depolarization of cells for a long period of time in mice medial prefrontal cortex. The elevation of excitatory/inhibitory balance provokes a deep impairment in information processing and social behavior dysfunction [15].

A magnetic resonance spectroscopy study has shown a reduced glutamate concentration in the striatum as compared with controls both in adults with idiopathic ASD and in mice models, especially in mutational model of SHANK3 and neuroligin–neurexin complex [16]. Similarly, reductions in GABA have been detected in magnetic resonance spectroscopy studies in subjects in an age-dependent manner, in motor, visual, auditory, somatosensory area, and in the perisylvian region of the left hemisphere, leading to abnormal information processing [17,18].

The plasma GABA and glutamate levels are altered in children with ASD. In particular, there is a significant elevation of plasma GABA and the glutamate/glutamine ratio while the levels of plasma glutamine and glutamate/GABA ratios are significantly lower as compared with the controls [19]. This imbalance between excitatory and inhibitory mechanisms in the GABA and glutamate neurophysiology has been linked to other neurodevelopmental disorders such as global developmental delay and mental retardation, schizophrenia, and epilepsy [20,21].

Atypical sensory perceptions are very common in ASD. Using magnetic resonance imaging (MRI) spectroscopy, Robertson et al. demonstrated a tight linkage between atypical dynamics of binocular rivalry in ASD and reduced GABAergic, but conserved glutamatergic levels, in the autistic occipital visual cortex [22].

MECP2 mutations provoke Rett syndrome and several neuropsychiatric disorders including autistic symptoms. Mutations in MeCP2 gene lead to a GABAergic dysfunction through reduced glutamic acid decarboxylase-1 and -2 levels and GABA immunoreactivity, changing the synaptic physiology and provoking numerous Rett syndrome and autistic-like characteristics, including repetitive behaviors in mice [23].

Some studies underlined an association between single-nucleotide polymorphisms (SNPs) of GABA receptors located in the chromosome 15q11–q13 of ASD subjects [24,25]. However, a recent meta-analysis has demonstrated that different SNPs of GABA receptor subunits B3, A5, and G3 had no correlations with autism in different ethnic populations [26].

The pharmacological approach with GABA modulators in autism aims to target the imbalance between excitatory glutamatergic and inhibitory GABAergic pathways. Arbaclofen, acamprosate, bumetanide, and valproate are the most studied substances. However, the majority of these studies are open-label trials and imply little statistical significance. A systematic review has, therefore, remarked that, to date, there is lack of evidence suggesting the use of GABA modulators to treat autism core symptoms, and further well-designed trials are needed [27].

## 4. Glutamate

Glutamate is the main excitatory neurotransmitter in the mammalian cortex. There are three main classes of receptors for glutamate, known as N-methyl-D-aspartate receptors (NMDARs), α-amino-3-hydroxy-5-methyl-4-isoxazolepropionic acid receptors (AMPARs), and metabotropic glutamate receptors [28]. Both NMDARs and AMPARs have also been implicated in ASD and much evidence supports this hypothesis [29]. 

Valproic acid-induced rodent models of autism have shown a selective overexpression of NR2A and NR2B subunits of NMDA receptors. This overexpression provoked enhanced NMDA receptor-mediated synaptic currents which led to an amplified postsynaptic plasticity in neocortical pyramidal neurons [30].

Modifications of AMPAR GluA2 subunit have deep effects on neuronal excitability and GluA2 dysregulation has been linked to different neuropsychiatric disorders such as intellectual disability and Rett syndrome [31]. Moreover, in a mouse model of CDKL5 deficiency disorder showing a phenotype characterized by autistic-like behaviors, intellectual disability, and seizures, a significant decrease in AMPAR GluA2 subunit in the hippocampus has been documented [32].

NMDA and NMDAR have also been correlated to autism. Specifically, mutations of GRIN2A and GRIN2B genes (respectively coding for GluN2A and GluN2B subunits) have been linked to ASD [33,34].

An alteration of NMDAR has been highlighted in several mouse models of autism such as Shank3 DC/DC mice, neuroligin-3 R451C knock-in mice, Fmr1−/− mice, and Shank2−/− mice [35,36]. 

Recent theories support the involvement of the cerebellum in ASD. Interestingly, for the first time, alterations have been demonstrated in the cerebellum granular layer of IB2 (islet brain-2) KO mouse models. The IB2 gene is implicated in Phelan–McDermid syndrome and provokes autistic symptoms and a severe motor delay. The IB2 KO mouse models have a NMDA receptor hyperactivity and hyperplasticity that determines an increase of excitatory/inhibitory balance and an enhanced long-term potentiation in mossy fiber and granule cells [37].

Nevertheless, an early correction of NMDAR dysfunction showed, in mouse models, a significant improvement in autistic-like behaviors [38,39,40].

Evidence shows that mutations in genes (SHANK, NLGN3, NLGN4, and UBE3A) involved in synapse formation and maintenance and in protein targeting are correlated both to the development of autistic traits and to glutamatergic dysregulation [41,42,43].

However, a large glutamatergic and GABAergic gene set analysis in subjects with ADHD and autism has shown only a significant association between glutamate gene set and hyperactivity and impulsivity symptom severity. No significant associations were found for autism symptoms in glutamate and GABA gene set which reinforces the need for further research on the genetics of excitatory/inhibitory imbalance in ASD [44].

In human beings, pharmacological enhancement or suppression of NMDAR function has determined an improvement in ASD symptoms [45]. In particular, an NMDAR agonist (D-cycloserine) significantly reduced social withdrawal and repetitive behavior [46,47]. Similarly, the administration of an NMDAR antagonist (memantine) improved stereotypies, lethargy, irritability, hyperactivity, and inattention suggesting a bidirectional NMDAR dysfunction [48,49].

## 5. Serotonin

Several studies have shown the involvement of the serotonin system in the etiology of autism during early brain development [50]. Serotonin (5-hydroxytryptamine, 5-HT) is a neurotransmitter belonging to the monoamine family; it is involved in the modulation of several developmental events, including cell division, cortical proliferation, migration, differentiation, cortical plasticity, and synaptogenesis [51,52]. Serotonin intervenes in various brain functions such as memory, learning ability, and has a role as a sleep and mood modulator [53,54]. Studies have revealed that the serotonin transporter (SERT or 5-HTT) or serotonin levels were higher in autistic children and in animal models as compared with controls, while there was postmortem evidence for reductions in both 5-HT2A and 5-HT1A binding in ASD brain [55,56,57]. Positron emission tomography (PET) studies revealed that healthy children between two and five years of age showed an elevated 5-HT synthesis, with a subsequent decline at puberty. Children with autism did not show this decline in the ability to synthesize serotonin over time, and the levels were significantly lower in these children at the age of two to five as compared with the controls, slightly increasing with age [58,59].

Polymorphisms in the SLC6A4 gene, which encodes for platelet and neuronal transport of 5-HT, have been associated with autism. These polymorphisms are functionally significant, and the higher 5-HT levels observed in ASD are substantial in children with SLC6A4 polymorphisms [60,61]. Differentially, there is animal evidence that embryos developing in Slc6a4^+/−^ dams have reduced resilience to prenatal stress which increases the offspring’s risk of developing ASD-like characteristics [62].

Several studies have detected platelet hyperserotonemia in ASD subjects with average increases of 20% to 50% [63,64,65]. Interestingly, this increase appeared to be specific to autism or ASD because it was not observed in intellectual disability or in other neuropsychiatric disorders [66,67].

Some selective serotonin reuptake inhibitors (SSRIs) have shown modest efficacy in the treatment of specific behaviors such as disruptive and repetitive symptoms; however, only fluoxetine has shown good evidence of decreasing global autism symptomatology [68,69].

All these studies confirm the role of serotonin in the pathophysiology of autism. However, the mechanism of elevation remains uncertain and the relationship with central serotonergic functioning needs further investigations.

## 6. Dopamine

In addition to being correlated to motor control, dopamine plays an important role in social cognition and behaviors especially through the mesocorticolimbic pathway [70].

Numerous studies have suggested that ASD could be linked to dopaminergic dysfunction and have hypothesized that dopamine imbalances in specific brain regions could lead to autistic behaviors [71]. In particular, autistic subjects have shown alterations in the mesocorticolimbic dopaminergic signaling pathway, such as reduced dopamine release in the prefrontal cortex and reduced neural response in the nucleus accumbens [72,73]. An article suggested that, in autism, social deficits were determined by a dysfunction of the mesocorticolimbic circuit, while the dysfunction of the nigrostriatal circuit led to stereotyped behaviors [74]. Concerning the nigrostriatal dopaminergic circuit, studies on mouse models have shown that drug-induced nigrostriatal pathway dysfunction caused stereotyped behaviors in mice [75]. Indeed, the administration of D1 dopaminergic receptor antagonists have reduced these behaviors [76]. 

A recent study supported the hypothesis that mesocorticolimbic circuit could impact social behaviors through the bidirectional control of dopaminergic projections from ventral tegmental area to nucleus accumbens. In particular, the optogenetic stimulation of dopaminergic ventral tegmental area neurons determined the activation of D1 receptors leading to an increase in the time that animals spent in social interaction, whereas inhibition had the opposite effect [77].

Genetic studies have shown an association between autism and several gene polymorphisms involved in dopaminergic pathways, such as dopamine receptors DR3 and DR4, or dopamine transporter (DAT) [78,79,80]. A recent study on mouse models highlighted that mutations in DAT provoked anomalous dopamine efflux and led to autistic-like behavioral phenotypes [81]. Dopaminergic gene polymorphisms should modulate emotion dysregulation and ADHD symptoms in children with ASD [82]. Moreover, haploinsufficiency of SHANK3 have reduced neuronal dopaminergic activity in the ventral tegmental area generating behavioral anomalies including impaired social skills [83].

Only the dopamine receptor blockers (risperidone and aripiprazole) are EMA/FDA-approved for the treatment of irritability, and they have also been shown to be effective in treating ASD repetitive behaviors [84,85].

All this evidence should lead to considering the administration of dopamine modulators as a therapeutic target for further studies in ASD behavioral treatment.

## 7. Acetylcholine 

Acetylcholine is the neurotransmitter used by motor neurons at the neuromuscular junction. It is also the main neurotransmitter of the parasympathetic nervous system and acts as a neurotransmitter and a neuromodulator in the CNS. 

The main evidence of cholinergic system abnormalities in ASD has included a significant reduction of nicotinic α4β2 subtype of ACh receptors (nAChRs) in the parietal and frontal cortex detected in post-mortem brain samples [86,87].

Another study has shown a reduction of cerebellar α4 nAChRs which could be linked to the loss of Purkinje cells and to a compensatory increase in α7 nAChRs [88]. 

Several studies on ASD animal models have shown the involvement of nAChRs in modulating social and repetitive behaviors [89].

There is animal evidence that α4β2 nAChRs are linked to autistic-like symptoms and the administration of ABT-418 (a neuronal nicotinic acetylcholine receptor agonist) determines a statistically significant improvement in these psychiatric symptoms [90].

Alpha4 nAChR subunit knock-out and beta2 nAChR subunit knock-out mice, respectively, show increases of anxiety and abnormal sleep pattern [91,92].

The α7 nicotinic receptor plays a particularly promising role in the pathogenesis of neuropsychiatric pathologies, including schizophrenia, ASD, ADHD, addictive disorders, because it is involved in sensory processing, cognition, working memory, attention and it is highly expressed in the regions involved in these cognitive functions such as hippocampus and frontal cortex [93,94].

Growing evidence supports the idea that the stimulation of α7 nAChR receptor has procognitive effects both in animal and in vivo models. These effects are mediated by PI3K/Akt signaling cascade crosstalk with the Wnt/-catenin signalling cascade and both transcriptional and non-transcriptional effects of catenin, metabolic effects of transient increases in the intraneuronal concentration of Ca^2+^, and changes in membrane potential [95].

Mutations involving the CHRNA7 gene in the chromosome region 15q13.3 have been correlated to autistic-like phenotypes [96]. In CHRNA7 null mutant mice, an increase of IL6 has been observed in mutant fetal brain due to maternal immune activation, and increased behavioral deficits in the offspring have also been observed. Moreover, it has been reported that the gestational choline supplementation improved the fetal brain’s response to maternal immune activation and prevented several induced behavioral abnormalities in the offspring [97].

Alpha7 and α4β2 subtypes of nAChRs are highly expressed in the CNS and, as we have observed, they are more involved in the ASD pathogenesis. Therefore, they have been detected as a possible therapeutic target. 

A randomized, double-blind, placebo-controlled trial on galantamine showed a statistical improvement in irritability, lethargy, and social withdrawal with good tolerability [98].

Evidence on animal models and on men has also demonstrated that donepezil has a good safety and tolerability profile and determined an improvement in behavioral dysfunctions [99].

3-(2,4-Dimethoxybenzylidene)-anabaseine (DMXB-A) is a selective partial agonist for α7- nAChRs and has shown its efficacy in a randomized, double-blind crossover trial on neurocognitive improvements in subjects with schizophrenia [100]. Analogous effects have been highlighted in two adult patients with ASD [101].

## 8. N-acetyl Aspartate

N-acetyl aspartate (NAA) is a widely diffuse metabolite in the human CNS. Its high brain concentration and its main functions remain uncertain. It is expressed in neurons, oligodendrocytes, and myelin and it is synthesized in the mitochondria derived from aspartic acid [102]. A decreased NAA concentration has been found in several psychiatric disorders and seemed to be correlated to a mitochondrial dysfunction [103]. The altered metabolic state can be reversed with psychopharmacological treatment capable of restoring a normal NAA level [104].

A functional MRI study has shown a significant reduction of NAA concentration in all brain regions and a specific reduction in the left frontal cortex as compared with the controls [105]. Another study showed the relationship between the dorsal striatal volume and NAA and glutamate levels in ASD and in obsessive-compulsive disorder as compared with the controls, underlying possible overlapping subcortical abnormalities [106]. Further studies are needed to elucidate the exact role of NAA in the pathogenesis of ASD.

## 9. Oxytocin and Arginine-Vasopressin 

In recent years, researchers have shown an increasing interest in oxytocin (OXT), another molecule that seems to be involved in the neurochemistry of autism. Arginine-vasopressin (AVP) belongs to the same superfamily as oxytocin. Their structure is very similar, and their genes are close on chromosome 20p13, separated only by 12 kb of DNA, and they have an opposite transcriptional orientation. For these reasons, two neuropeptides influencing one another’s functions have effects on the same neural structures in the central and autonomic nervous systems and they both modulate human behavior [107].

Oxytocin is a neuropeptide involved in a number of physiological processes, including parturition and lactation. It has been shown to modify synaptic plasticity and to modulate social behaviors such as eye contact, social recognition, aggressivity, sociosexual behaviors with a role in sculpting emotional and social “self” [108,109].

Altered plasma levels of OXT and AVP have been reported in autistic individuals and they are often correlated to aberrant functional connectivity [110,111]. An association between the degree of methylation of the oxytocin receptor (OXTR) gene and autistic symptoms has been reported [112,113]. However, the results are not all consistent.

The genetics of OXT and AVP receptors has been widely explored. In particular, two microsatellites, RS1 and RS3, within the promoter region of vasopressin receptor 1A (V1AR), and rs28632197 and rs35369693 SNPs of vasopressin V1b receptor (V1BR), have been found to be significantly linked to ASD. Two SNPs of the OXT receptor gene (rs53576 and rs225429) are also significantly associated with ASD. However, the mechanisms whereby these polymorphisms contribute to ASD pathogenesis have yet to be completely clarified [114,115].

Interestingly, mutations in SHANK3 (a postsynaptic scaffolding protein implicated in synapse development and ASDs) [116] have effects on the oxytocinergic system and this alteration could be the cause of some behavioral phenotypes related to synaptic plasticity in autism [117]. Recently, it has been hypothesized that the failure of the oxytocinergic system during the early stages of neurodevelopment could affect social behavior by altering synaptic activity and plasticity [118].

Animal model research has documented that the administration of OXT and AVP was able to rescue autistic traits and increase social skills [119,120,121]. In humans, there is some evidence that the administration of oxytocin reduces some dysfunctional behaviors associated with autism, especially social skills, repetitive behaviors, anxiety, irritability, and self-injurious behaviors [122,123,124]. However, a recent meta-analysis that reviewed randomized controlled trials on ASD symptomatology did reveal that there was no benefit of oxytocin over placebo and provided further proof to support existing evidence [125].

In the end, there is evidence for crucial interaction between the dopaminergic, AXT-AVP, and serotoninergic systems in different areas of social brain with great influence on human social behavior [126].

## 10. Melatonin 

Children with ASD often have sleep disorders, such as difficulty falling and staying asleep, and parasomnia [127,128]. Melatonin is a major regulator of the sleep-wake rhythm, reduces sleep latency, it is a powerful antioxidant, it has a role in neurodevelopment and plasticity, and it can be important in placental homeostasis [129] and immunity [130]. Studies involving autistic individuals show lower melatonin or melatonin metabolite plasma levels and lower urinary melatonin sulfate excretion rates [131]. Moreover, a recent research showed that 6-sulfatoxymelatonin levels were significantly lower in mothers with an ASD child than in the controls [132]. During pregnancy, melatonin is able to cross the placenta providing photoperiodic information to the fetus and establishing a normal sleep cycle which is essential for normal neurodevelopment. Indeed, maternal melatonin, before the maturation of the fetal pineal gland, protects against brain inflammation and injury [133].

Some studies underline that specific gene abnormalities (MTNR1A, MTNR1B, GPR50, and ASMT) could contribute to reduced melatonin level or to altered melatonin receptor function or they could be involved in melatonin synthesis in a small percentage of ASD patients. [134] 

The use of melatonin for the treatment of chronic sleep-wake cycle disorders in children with autism is constantly increasing [135]. Melatonin, associated with educational and behavioral interventions, appears to be the most effective treatment for improving both sleep problems and daytime behaviors [136]. Indeed, its effects go beyond sleep, melatonin acts on anxiety, depression, pain and gastrointestinal dysfunctions, improving the well-being of ASD subjects [137]. 

## 11. Vitamin D 

Vitamin D is an active steroid with an important role in antioxidant activity, neuronal calcium regulation, immunomodulation, and in the regulation of neurotransmitters and neurotrophic factors [138]. In the brain, vitamin D also plays a role in neuronal proliferation and in synaptic plasticity [139]. Lower levels of vitamin D have been reported in autistic individuals [140,141]. There is animal evidence that a partial lack of vitamin D exposes the brain to neuroinflammation and that the exogenous supplementation has a protective effect ameliorating neurotoxicity, inflammation, and DNA damage [142]. Vitamin D has an important role in neurotransmission. In fact, it regulates glutamate, GABA, serotonin, dopamine, and it alters immune function relevant in ASD pathogenesis and causes steroid and placental dysregulation [143].

Moreover, it seems that 25-hydroxyvitamin D deficiency either at mid-gestation or at birth is associated with an increase in autistic traits in children [144].

Exogenous vitamin D supplementation can have beneficial effects in ASD children and improve signs and symptoms of ASD [145,146], and the American Academy of Pediatrics recommends vitamin D supplementation during infancy and childhood [147]. However, a recent randomized placebo-controlled trial underlined that vitamin D supplementation had limited beneficial effects on children with ASD without any effect on the primary outcome [148].

## 12. Orexin System

Recently, the interest of researchers with respect to orexin related to autism has been increasing. Orexin, also known as hypocretin, is a neuropeptide secreted by neurons located in the lateral hypothalamus and perifornical areas. Orexergic fibers have great distribution in the brain and they have many physiological functions, such as excitement, sleep regulation, cognition, stress, appetite and metabolism [149]. Orexin dysfunction appears to be related to various neurological disorders including addiction, depression, anxiety, and schizophrenia [150]. Considering the prevalence of sleep disorders in individuals with ASD [114,151], it is possible to hypothesize that an alteration of the orexinergic system could be implicated in the pathogenesis of these disorders. Sleep disturbances in ASD patients depend on the increased orexinergic system activity (probably due to amygdala dysfunction) associated with a reduced serotoninergic and melatoninergic system activity [152,153]. In a recent study, plasma levels of Orexin A were higher in ASD patients than in the healthy population [146]. Another study underlined a strong correlation between plasma levels of oxytocin and orexin A in the ASD groups investigated. This finding supported the contribution of oxytocinergic mechanisms in ASD [154]. However, there are few studies in this field and the results are still contradictory.

## 13. Endogenous Opioids

Endogenous opioids are peptides that act as neuromodulators in the CNS. There are three types of endogenous opioids, i.e., beta-endorphins, enkephalins and dynorphins. Opioid administration determines behavioral effects, such as insensitivity to pain, affective lability, stereotypical behavior, and reduced socialization [155,156]. Blood and liquor studies have shown that beta-endorphin levels are altered in subjects with ASD, although not all results have been concordant [157,158]. Interestingly, a study has shown elevated plasma levels of beta-endorphin in ASD individuals after venipuncture, suggesting a different mode of pain expression and a lower ability to regulate their emotional response [159].

Some studies have examined the effects of opioid antagonists (naloxone and naltrexone) in individuals with autism and a reduction in self-aggressive behavior, hyperactivity, restlessness, and withdrawal has been demonstrated [160,161]. However, a recent systematic review clarified that although opioids can improve hyperactivity and restlessness in ASD children, there was not sufficient evidence that they had an impact on the core symptoms of autism in the majority of participants. Nevertheless, a subgroup of children with autism and abnormal endorphin levels have responded to naltrexone [162].

## 14. Discussion

ASD is a complex neurobehavioral syndrome and no specific causes have yet been identified. Anatomical brain abnormalities, genetic anomalies, and neurochemical dysfunctions of various neurotransmitters and neuropeptides including GABA and glutamate, serotonin, dopamine, N-acetyl aspartate, oxytocin, arginine-vasopressin, melatonin, vitamin D, orexin, opioids, and acetylcholine contribute to the onset of autism. Our review suggests the important role played by altered neurotransmission in the etiopathogenesis of ASD (see Table 1). It is clear that many pathways are involved in determining autism, but how these biological systems interact with each other remains obscure. Further neurochemical network studies on early neurodevelopment alterations are required. Advancing the understanding of the etiology of the ASD mechanisms represents a real challenge, mainly due to the enormous heterogeneity of ASD.

The evidence gathered by our review supports the existence of several dysregulated neurotransmitters and neuropeptides in animal models and in patients with autism. Although there is some evidence suggesting that specific receptor anomalies lead to specific phenotypic variations it is very hard to highlight the pathogenetic role of every neuronal receptor in determining the autism phenotype. 

In this framework, some clinical features of ASD could, at least partially, find an explanation such as sensory integration alteration [163], and neuropsychological and psychological dysfunction [164,165,166]. 

The heterogeneity of autism makes it very difficult to detect exclusive neurobiological and genetic traits of ASD and, to date, there are only a very few replicated neurochemical findings.

Additional efforts are required to understand whether these anomalies have a primary etiological role or whether they are rather secondary epiphenomena of a global cerebral dysfunction. Most studies about autism etiology more fruitfully examine specific domains of behavior or single impairments rather than the whole autism phenotype. 

Advances in research could lead to new therapeutic strategies that could be useful for improving and perhaps even preventing autism symptoms. 

The growing body of evidence reported by our review on the pharmacological substances targeting receptor abnormalities often showed conflicting results and we hope that higher quality studies would be conducted in order to clarify what receptor system could represent an effective pharmacological target for the treatment of autism symptomatology.

In conclusion, additional evidence on the neurochemical alteration of autism is needed and a greater knowledge in this field could lead to a completely new approach to the pharmacological management of autism and to the identification of biomarkers with greater specificity and sensitivity.

## Figures and Tables

**Table 1 brainsci-10-00163-t001:** Main neurochemical findings.

Molecule	Imbalance	Genes	Animal Models	Pharmacological Approach
**GABA**	↓ motor, visual, auditory, somatosensory cortex ↑ blood	MECP2 GABRA5, GABRG3, GABRB3 *	Viaat-Cre mice^+^	Arbaclofen, acamprosate, bumetanide, and valproate
**Glutamate**	↓ striatum ↑ blood	SHANK, NLGN3, NLGN4, UBE3A, GRIN2A, GRIN2B, CDKL5	Nlgn3 KO mice Shank3 KO mice Shank2 KO mice VPA-mice Cdkl5 KO mice IB2 KO mice	D-cycloserine Memantine Amantadine mGLuR5-antagonists
**Serotonin**	↑ brain and blood; ↓ 5-HT2A, 5-HT1A binding	SLC6A4	SERT Ala56 mice Slc6a4 +/− mice	Selective serotonin reuptake inhibitor
**Dopamine**	↓ prefrontal cortex Dysregulation of mesocorticolimbic and nigrostriatal circuit	SLC6A3 SHANK3 DRD3 DRD4	Stereotypic deer mice DAT T356M^+/−^	Dopamine receptor blockers
**Acetylcholine**	↓ α4β2 nAChRs in parietal and frontal cortex ↓ α4 ↑ α7 nAChRs in cerebellum	CHRNA7 CHRNA4 CHRNB2	CHRNA7 null mutant miceb TBR mouse PTZ-kindled mice	ABT-418 α7 nAChR modulators galantamine donepezil
**Oxytocin and arginine-vasopressin**	↑ OXT plasma	OXTR AVPR1A, AVPR1B Shank3	OXTR KO mice V1aR knock-out mice	Oxytocin
**Melatonin**	↓ plasma ↓ urinary excretion	MTNR1A, MTNR1B GPR50 ASMT	MT1 and MT2 receptor knock-out mice	Melatonin

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
