# Peer review of "The Neurochemistry of Autism"

_brainsci, 2020, doi:10.3390/brainsci10030163_

Round 1

Reviewer 1 Report

The manuscript by Marotta et al. summarizes a good amount of literature about the neurochemistry of autism spectrum disorders. I appreciated the efforts to cover as much literature as possible in the space allowed by an Opinion type of article. Nevertheless, I believe that the authors might extend the section on glutamatergic receptors. Almost all the main theories about ASDs pathophysiology consider, to some extent, the alteration of the excitatory/inhibitory balance, usually due to alterations in the GABAergic and glutamatergic signaling. Therefore, it seems reasonable to divide the paragraphs on GABA and glutamate, which I think deserve separate and thorough descriptions (see below for more details). English needs to be significantly improved throughout the whole text.

This manuscript provides a detailed review that has the advantage of bringing together the evidence from the main researches on different neurochemical factors in ASDs.

Major comments:

1) English needs to be significantly improved (e.g., sentences at lines 21, 224, 249, 269, 325, 342, 373; but all the manuscript should be checked).

2) GABA and glutamate should be addressed in separate paragraphs.

3) In the glutamate paragraph, more attention should be devoted to NMDARs. A relevant theory of ASD pathophysiology is related to NMDARs functional alteration in ASD models, as the valproic-acid insult-based model used by Markram and colleagues (see Rinaldi et al., 2007). Moreover, a recent model of the Phelan-McDermid syndrome is the IB2 KO model, which shows increased NMDAR-mediated currents in the cerebellum (Soda et al., 2019). Since the importance of this receptor in ASD theories, I believe it should be treated more in-depth in this manuscript.

4) The statement (lines 389-392): “The growing body of evidence reported by our overview […] in order to clarify what receptor system could represent an effective pharmacological target for the treatment of autism symptomatology”. As it is, it seems that one neurotransmitter system might arise as the ideal target for all autism syndromes. This seems unlikely, even more considering the amount of literature and experimental evidence on the different molecules described in this manuscript. I think it is not reasonable to expect that one factor will be determinant over the others in any scenario. If this is not the real intention of the authors, I suggest to re-phrase that sentence.

Minor comments:

a) The acronyms should be explained when first used (for example, ADHD is explained at line 220 though cited before multiple times), and then used consistently (“autism spectrum disorders” is used in extenso at line 338 while being already defined at line 46).

b) The reference style should be checked for consistency [e.g., line 194, Ref (38)].

c) For the same Ref (38): is it cited correctly in the Dopamine section? It seems to be related to NMDARs in ASD.

d) line 235: not sure that alpha7 should be written as A7, though at the beginning of a sentence.

e) line 224: this sentence has no verb (maybe “supporting” should be “supports”?)

f) line 249: the use of the term “revisable” seems a mistake (but all the manuscript should be checked for English, as already pointed out before).

g) The manuscript could take advantage of a table to summarize the main models and dysfunctions for every molecule considered. This suggestion is included in the minor issues, since it could be trickier than it seems, given the amount of evidence reported here. The authors are free to decide whether and how to produce said table.

Author Response

I would like to thank the editor and the reviewers for their advice to the article. As you requested, we have made all the necessary changes in our manuscript to address your concerns.

In particular we made the following main changes:

  • A complete revision of the English language of the whole text
  • GABA and glutamate have been analysed in separate paragraphs
  • More attention was addressed to NMDARs functional alteration
  • more details have been provided about the mesocorticolimbic circuit in social behaviors
  • We added a table summarizing the main neurochemical findings

Thank you for giving us these useful suggestions to improve the paper. We are sure that, thanks to your suggestions, the manuscript has greatly strengthened in this revised form. If there are any further questions, please feel free to let me know.

Sincerely,

Luigi Vetri, MD

Department of Sciences for Health Promotion and Mother and Child Care “G. D'Alessandro,”

University of Palermo

Address: via Del Vespro 129, 90127 Palermo, Italy

Phone: +39 3286434126

Email: luigi.vetri@gmail.com

Reviewer 2 Report

The authors have done a very good job of appraising the literature, as it relates to the role of these select neurological circuits and ASD. However, the review would be strengthened if the authors too their information a step further and elaborated on how alterations to these specific circuits could impact certain features of ASD. The authors have started to do this in many places, but additional details and evidence needs to be provided to create a stronger rationale for the importance of such neural circuitry changes in ASD. For example, there is a wealth of published information that links alterations in the dopamine circuit to social cognition. The authors make a statement and provide a citation, to this point, but more detail should be provided about these findings and how alteration in the mesocorticolimbic circuit could impact social behaviors.

These comments about detail and additional evidence should be applied to all sections of this review.

Author Response

Dear reviewer,

I would like to thank the editor and the reviewers for their advice to the article. As you requested, we have made all the necessary changes in our manuscript to address your concerns.

In particular we made the following main changes:

-        A complete revision of the English language of the whole text

-        GABA and glutamate have been analysed in separate paragraphs

-        More attention was addressed to NMDARs functional alteration

-        We added a table summarizing the main neurochemical findings

-        more details have been provided about the mesocorticolimbic circuit in social behaviors:

for example, you can find additional evidence about GABA/glutamate (from line 116), dopamine (from line 228), OXT-AVP and serotoninergic systems (from line 347).

And we add further references about social cognition that we hope you can appreciate such us:

Skuse, D. H., & Gallagher, L. (2009). Dopaminergic-neuropeptide interactions in the social brain. Trends in cognitive sciences, 13(1), 27-35.

Gunaydin, L. A., Grosenick, L., Finkelstein, J. C., Kauvar, I. V., Fenno, L. E., Adhikari, A., ... & Tye, K. M. (2014). Natural neural projection dynamics underlying social behavior. Cell, 157(7), 1535-1551.

Tidey, J. W., & Miczek, K. A. (1996). Social defeat stress selectively alters mesocorticolimbic dopamine release: an in vivo microdialysis study. Brain research, 721(1-2), 140-149.

Yizhar, O., Fenno, L. E., Prigge, M., Schneider, F., Davidson, T. J., O’shea, D. J., ... & Stehfest, K. (2011). Neocortical excitation/inhibition balance in information processing and social dysfunction. Nature, 477(7363), 171-178.

Thank you for giving us these useful suggestions to improve the paper. We are sure that, thanks to your suggestions, the manuscript has greatly strengthened in this revised form. If there are any further questions, please feel free to let me know.

Sincerely,

Luigi Vetri, MD

Department of Sciences for Health Promotion and Mother and Child Care “G. D'Alessandro,”

University of Palermo

Address: via Del Vespro 129, 90127 Palermo, Italy

Phone: +39 3286434126

Email: luigi.vetri@gmail.com

Round 2

Reviewer 2 Report

The authors have done a very nice job of addressing the previous comments. No additional concerns are expressed.